# Typology of Parent-to-Child Emotions: A Study of Japanese Parents of a Foetus up to a 12-Year-Old Child

**DOI:** 10.3390/healthcare12090881

**Published:** 2024-04-24

**Authors:** Ayako Hada, Yukiko Ohashi, Yuriko Usui, Toshinori Kitamura

**Affiliations:** 1Kitamura Institute of Mental Health Tokyo, Tokyo 151-0063, Japan; y-ohashi@jiu.ac.jp (Y.O.); yusui@g.ecc.u-tokyo.ac.jp (Y.U.); kitamura@institute-of-mental-health.jp (T.K.); 2Kitamura KOKORO Clinic Mental Health, Tokyo 151-0063, Japan; 3Department of Community Mental Health & Law, National Institute of Mental Health, National Center of Neurology and Psychiatry, Tokyo 187-8553, Japan; 4Department of Mental Health and Psychiatric Nursing, Tokyo Medical and Dental University, Tokyo 113-8510, Japan; 5Faculty of Nursing, Josai International University, Togane 283-8555, Japan; 6Department of Midwifery and Women’s Health, Division of Health Sciences and Nursing, Graduate School of Medicine, The University of Tokyo, Tokyo 113-0033, Japan; 7T. and F. Kitamura Foundation for Studies and Skill Advancement in Mental Health, Tokyo 151-0063, Japan

**Keywords:** cluster analysis, parent-to-child emotions, basic emotions, self-conscious emotions, parental gender, child gender, child age

## Abstract

**Background:** Emotions are the fundamental origin of parent–child bonding, which is measurable by the Scale for Parent-to-Child Emotions (SPCE) based on the theories of basic and self-conscious emotions. **Methods:** This study is based on the data from a cross-sectional study that we previously reported. The data consist of fathers and mothers who had a child/children, whose eldest child’s age was at the foetal stage up to 12 years old, and were recruited via the Internet (N = 4600). A series of cluster analyses using factor scores (theta[*Ө*]s) of all domains of the SPCE were conducted. After the clusters emerged, the fathers and mothers allocated to each cluster were compared by the child’s age stage. The validation of the classifications was also conducted using ANOVAs and chi-squared tests. A discriminant function analysis was conducted. **Results:** The participant mothers and fathers were classified into Cluster 1 (Lack of Bonding Emotions, *n* = 509), Cluster 2 (Bonding Disorder, *n* = 1471), Cluster 3 (Ambivalent Bonding Emotions, *n* = 1211), and Cluster 4 (Positive Bonding, *n* = 1409). Across the four clusters, there were no differences in the age of the parents or the gender of the child. During the second trimester, mothers made up the majority of Cluster 4 (Positive Bonding), totalling 81 cases (37.5%), whereas fathers made up the majority of Cluster 2 (Bonding Disorder), totalling 126 cases (60.0%). The three linear discriminants (LDs) well predicted the four clusters, and their functions showed cross validation. **Conclusions:** The typology of the SPCE is helpful to understand individual differences in terms of parental emotional bonding.

## 1. Introduction

Emotions are the fundamental origin of parent–child bonding which develop via the interaction of the parent–child dyad. These emotions are differentiated into a variety of emotion categories. The primary function of emotions is to mobilise the organism to deal quickly with important interpersonal encounters and to help determine what types of actions are adaptable in interpersonal relationships [1]. This is the case for parent-to-child emotions. A parent feels various emotions that motivate adaptive behaviours in the interaction of the parent–child dyad. For example, a higher quality of maternal bonding was correlated with a higher level of the child’s attachment to the parent, lower parent-reported colic rating, easier temperament, and positive infant mood [2]. The emotions a parent feels towards a child play an important role in the quality of their interaction. Therefore, it is meaningful to clarify what kind of emotions parents feel within their interactions with their child.

A different variety of emotions may be divided into two types: basic emotions and self-conscious emotions. Basic emotions and self-conscious emotions are underlying in humans and the key rubrics for parent-to-child emotions [3,4]. Basic emotions are crucial in evolution and adaptation for survival [5]. Therefore, in parent–child relationships, which need to be kept secure, basic emotions are essential. Self-conscious emotions bring the interpretation of interpersonal events and motivate a person to take appropriate actions in terms of morals [6]. Hence, parents’ self-conscious emotions elicit appropriate actions as a parent in terms of morals in parent–child relationships. Although the terms ‘attachment’ and ‘bonding’ are sometimes used interchangeably, they should be distinguished. The term ‘bonding’ has the direction of a parent to a child/foetus. On the other hand, the term ‘attachment’ has the direction of a child/foetus to a parent. Walsh [7] pointed out that the terminology of attachment is not accurate in describing how parents feel. Furthermore, Kinsey and Hupcey [8] proposed that *the affective state of the mother; maternal feelings and emotions towards the infant are the primary indicator of maternal-infant bonding*. 

Basic emotions (i.e., happiness, anger, fear, sadness, disgust, and surprise) are expressed via facial expressions and convey different classes of information: antecedents, thoughts, an initial state, a metaphor, what the expresser is likely to do next, what the expresser wants the perceiver to do, or an emotive word [9]. Basic emotions are important communication tools among humans. This is no less important in parent–child dyads. Children and adults regulate their emotional states and communicate affectively [10]. Even newborns can discriminate between happy and sad, happy and surprised, and sad and surprised facial expressions [11]. Three- to five-month-old children can discriminate among the happy, angry, fearful, and sad expressions of others [12,13]. An experimental study using a continuous recording of electrical signals demonstrated that seven-month-old infants could discriminate between sad and happy expressions [14]. Infants attain a capacity to perceive, interpret, and respond to other people’s positive and negative facial expressions in the first two years of life [15]. Thus, the ability to perceive and understand discrete facial expressions develops in early human life. Infants receive important information from their parents via the emotional expressions in their emotional interactions, which helps their development.

Self-conscious emotions are less prototypical facial expressions than basic emotions. However, shame, guilt, and pride can be identified by employing a combination of physical characteristics [16,17,18,19]. Whereas appraisals of internal attributions for failure tend to elicit negative self-conscious emotions (e.g., shame or guilt), appraisals of internal attributions for success tend to produce positive self-conscious emotions (e.g., alpha pride or beta pride) [20]. Self-conscious emotions facilitate interpersonal relationships with reciprocal altruism [21,22]. 

Shame and guilt are interpersonal emotions, in that they are most likely to arise in relationships with others [20]. Shame and guilt have important implications for consequent motivation and interpersonal functions, but shame and guilt have different pathways to interpersonal behaviours. Guilt in response to clear transgressions is generally unrelated to psychological problems, whereas shame is associated with a wide range of psychological maladjustments [23] including aggression [24], somatisation, obsession and compulsion, psychoticism, paranoid ideation, hostility, interpersonal sensitivity, anxiety, and depression [25]. On the other hand, guilt proneness is correlated with empathy: an empathic person is more likely to feel guilt than a non-empathic person [26].

Pride has the functions of strengthening and motivating socially valued behaviours that help maintain a positive self-concept and others’ respect. Prideful expression leads a person to respond adaptively to goals within social values and increase their social status [19,27]. Prideful emotions motivate us to evaluate and behave following moral standards and to avoid impulsion for immoral behaviour by rewarding and reinforcing one’s commitment to the ethics of autonomy, community, and divinity [6]. Thus, pride involves morality with the internal self and plays an important role in the interpersonal function. Pride, however, has two aspects: alpha pride and beta pride. Alpha pride is hubristic pride, whereas beta pride is authentic pride. Shame and alpha pride are associated with low well-being, whereas guilt and beta pride are linked to high well-being [28].

Self-conscious emotions are important not only in adult–adult interactions but also between a parent and child. Parental behaviours are likely to be influenced by self-conscious emotions. Self-conscious emotions elicit substantial motivations and ensure parenting behaviours. Parental shame can predict maladaptive parenting behaviours, whereas parental guilt can predict adaptive ones when a child did something wrong [29]. Social applause results in pride, which in turn motivates behaviours aimed at maintaining it [6,18,30]. Williams and DeSteno [30] pointed out that individuals often feel pride when their child succeeds. In this vein, self-conscious emotions are crucial emotions inside the self as a parent. When one encounters events relevant to his/her own child, he/she evaluates him/herself on whether he/she is or is becoming an ideal parent by referring to the parental norm attained through his/her own experience. Evaluations relating to violations of the parental norm make him/her feel ashamed or guilty. On the contrary, evaluating the self to believe and obey the parental norm makes him/her feel alpha and beta prides. The emotional responses of parents inside self-motivate them to behave according to their parental norms. Thus, self-conscious emotions are key emotions which shape the internal figure as a parent, which is crucial for parent-to-child emotions. 

The Scale for Parent-to-Child Emotions (SPCE [4]) is a unique scale used to evaluate parent-to-child emotions, covering basic and self-conscious emotions as parental emotional bonding. In the SPCE, basic emotion domains include happiness, anger, fear, sadness, and disgust, and self-conscious emotion domains include shame, guilt, and alpha and beta prides. Studies based on the classical test theory (CTT) and item response theory (IRT) showed the scale’s robust construction and the reliable characteristics of the scale items to measure latent traits. The SPCE has high versatility and can measure parent-to-child emotions both across different genders of parents and the different age stages of their child. The SPCE may facilitate several clinical or research questions regarding parental emotional bonding. The SPCE is likely to make individual differences in parental emotional bonding more apparent.

What remains to be answered is how many types (groups) parents can be categorised into in terms of parental emotional bonding states: what kinds of characteristics identify parents of these groups of parental emotional bonding? The latter includes the following: Do fathers and mothers differ in such categories? Can we estimate groups from the SPCE scores? Answers to these questions could be useful for clinical interventions and lead to further research on perinatal bonding. A cluster analysis using partitioning around medoids (PAM) can classify all cases in our data into several types (groups), with parental emotional bonding measured by the SPCE. We expect that the characteristics of these types (groups) will emerge, and the probability of belonging to a given cluster can be predicted by conducting a linear discriminant analysis (LDA).

Cluster analysis is the art of finding types (groups) in data. Classifying similar objects into several groups is useful for decisions and subsequent actions in clinical situations. There are many classification methods in cluster analysis, and they have advantages and disadvantages. Partitioning around medoids (PAM) is based on finding the *k* representative objects (medoids) among the objects in the data set. After finding a set of *k* medoids, each object from the data set is assigned to the nearest medoid, and then, the *k* clusters are constructed [31]. The goal of the algorithm is to minimise the average dissimilarity of the objects to their closest selected object. PAM is the popular algorithm for clustering. With PAM clustering, we can determine the number of clusters a priori. Therefore, the quality of clusters is compared to assess how many clusters are optimal for the data set. The quality of the clusters can be measured by silhouette widths. A silhouette width is a measure of how close an observation is to other observations in the same cluster compared to its proximity to observations in the neighbouring cluster. A silhouette width of 1 means that clusters are well defined, whereas a silhouette width of 0 means that clusters are highly overlapping. The average silhouette width evaluates the clustering validity and may be used to select an appropriate number of clusters [32].

A discriminant analysis (DA) is a multivariate statistic method used to separate two or more groups based on the scores derived from an appropriate *statistical decision function* that consists of one or more continuous predictor variables [33]. A linear discriminant analysis (LDA), which is one of the methods of DA, is a well-known method for feature extraction including statistical pattern recognition. An LDA is used to predict the probability of belonging to a given cluster (or category/class) based on predictor variables. This statistical method was introduced by Fisher (1936) for two classes [34], and then Rao [35] generalised it to multiple classes. A classical LDA is important in statistical pattern recognition. For a data set containing *k* clusters, the classical solution to an LDA extracts at most *k* − 1 features. We can find directions, called linear discriminants (LDs), that maximise the separation between clusters using an LDA. These directions are linear combinations of predictor variables. The probabilities of class memberships corresponding to observation can be predicted. The LDA is a simple, linear, supervised maximum likelihood (ML) algorithm used for prediction. Based on Bayes theorem, an LDA estimates the probability that a new object belongs to a certain class. An LDA has a high accuracy, sensitivity, specificity, and discriminant power [36]. 

## 2. Methods

### 2.1. Study Procedure and Participants

The data used in this report came from the data set of a cross-sectional study that we previously reported [4]. The target of this study was fathers and mothers who had a child/children, whose eldest child’s age was at the foetal stage up to 12 years old. Our only exclusion criterion was parents who had little command of Japanese. We asked participants to respond to the questionnaire, focussing on the eldest (or only) child (including a foetus). Twenty segments were created to allocate all participants by the parent’s gender (father/mother) and the child’s age stages. We aimed at recruiting 250 participants for each of the 20 segments: two parental genders (fathers and mothers) × ten age ranges of children (including foetuses). Those age ranges were as follows: (a) 1st trimester in foetal stage, (b) 2nd trimester in foetal stage, (c) 3rd trimester in foetal stage, (d) 0 to 1 month old, (e) 2 to 6 months old, (f) 7 to 17 months old, (g) 18 months to 2 years old, (h) 3 to 5 years old, (i) 6 to 8 years old, and (j) 9 to 12 years old. 

Parents were recruited from 47 prefectures in Japan with the cooperation of Rakuten Insight Inc. (Setagaya, Tokyo), which has research panels to recruit parents who were or whose partners were pregnant or lived with their 0- to 12-year-old child/children. However, 250 participants were not available for four segments: 1st trimester in foetal stage and 2nd trimester in foetal stage, for both parental genders. As a result, a total of 4600 parents responded. The demographics of this sample were reported elsewhere [4]. A web survey platform was created by Rakuten Insight Inc. All of the necessary information for participation, i.e., the aims of this research, affiliations of the principal researcher, and information about ethical considerations, were contained in the survey platform. The web page of our survey was available from 30 November to 6 December 2021.

### 2.2. Measurement

Parental emotional bonding: Parental emotional bonding was measured using the Scale of Parent-to-Child Emotions (SPCE [4]). The SPCE was developed to measure parent-to-child emotions based on the theoretical background of basic emotions and self-conscious emotions. The SPCE consists of 5 basic emotion domains (happiness [4 items], anger [6 items], fear [4 items], sadness [5 items], and disgust [5 items]) and 4 self-conscious emotion domains (shame [5 items], guilt [7 items], alpha pride [3 items], and beta pride [4 items]). All items have a 7-point rating scale. Measurement invariance was accepted across parents’ genders as well as different child’s ages: foetal stage, infant preschool age, and school age. An item analysis using IRT was also performed, and items with a flagged differential item functioning (DIF) were removed in the procedure of scale development. We used factor scores (theta[*Ө*]s) in all domains of the SPCE for cluster analysis in this study.

### 2.3. Statistical Analyses

*Cluster analysis*: We hypothesised that 3 or 4 cluster numbers would be appropriate for clinical settings. We therefore performed a cluster analysis, with the cases classified into 1, 2, 3, 4, and 5 clusters, using PAM clustering methods. A series of silhouette analyses were conducted to determine the most appropriate cluster numbers. As we mentioned previously, basic and self-conscious emotions are the origins which formulate parental emotional bonding. Therefore, *Ө* s in the IRT of all domains of the SPCE (i.e., happiness, anger, fear, sadness, disgust, shame, guilt, alpha pride, and beta pride) were used for PAM clustering. 

*Validation of classifications*: After the selection of the best classification, we performed a one-way analysis of variance (ANOVA) and Tukey post hoc comparisons for the scores of the SPCE domains and the parents’ and their child’s genders. The cases of the parents’ gender and their child’s gender by each cluster were counted, and chi-squared tests were performed. Furthermore, we counted cases belonging to each cluster by each segment, and the number of cases was compared with the parents’ gender. Among the same range of a child’s age, chi-squared tests between fathers and mothers were performed. In addition, the number of cases belonging to each cluster by each group of combinations of parents’ gender and their child’s gender was counted. Because of multiple comparisons, we set the significant level at 0.1% (*p* < 0.001).

*Discriminant function analysis*: The samples were divided into two groups: train data (*n* = 2300) and test data (*n* = 2300). Train data are for the LDA, and test data are for validating the LDA. The whole sample was divided into two groups by odd–even numbers in order to assure homogeneity. Then, we performed an LDA to find the directions that maximised the separation between clusters (following the results of the cluster analysis) using the train data. To examine the accuracy of the LDA, the number of cases belonging to a predicted cluster based on the LDA and true (observed) cluster, sensitivity, specificity, positive predictive value (PPV), and negative predictive value (NPV) for both the train and test data were calculated. 

All statistical analyses were performed using R (4.1.2). Theta scores for all SPCE domains were calculated by using the R package ‘ltm’, version 1.2-0 [37], the cluster analysis was performed using the R package ‘cluster’, version 2.1.4 [38], the ANOVA was performed using the R package ‘car’, version 3.1-1 [39], and the LDA was performed using the R package ‘MASS’, version 7.3–58.2 [40].

## 3. Results

*Cluster analysis*: A series of cluster analyses were performed to classify cases into a number of cluster(s) *k* = 1, 2, 3, 4, and 5 (Table 1). The best mean silhouette width was 0.38 (*k* = 2). The second-best mean silhouette width was 0.28 (*k* = 4). A four-cluster solution was chosen based on the silhouette index, practical considerations (such as the sample size), and interpretability. All cases were classified into Cluster 1 (*n* = 509), Cluster 2 (*n* = 1471), Cluster 3 (*n* = 1211), and Cluster 4 (*n* = 1409).

*Validation of classifications*: All mean *Ө* s showed significant differences (*p* < 0.001). The effect sizes are large (*η*^2^ = 0.38 to 0.79). (Table 2). The first cluster consisted of 509 (11%) parents. They were characterised by the lowest scores across all the SPCE domains. The second cluster consisted of 1471 parents (32%) characterised by high negative emotions (i.e., anger, fear, sadness, disgust, shame, and guilt) and low positive emotions (i.e., happiness, alpha pride, and beta pride). The third cluster consisted of 1211 parents (26%). They were characterised by the middle ranges across all domains of the SPCE. Finally, the fourth cluster consisted of 1409 parents (31%). They were characterised by high positive emotions and low negative emotions (Figure 1). Clusters 1, 2, 3, and 4 were named Lack of Bonding Emotions, Bonding Disorder, Ambivalent Bonding, and Positive Bonding, respectively. 

The parents’ mean age comparisons between the four clusters showed that Cluster 2 was significantly higher than Clusters 1 and 4 (*p* < 0.001) but not significant in comparison with Cluster 3 (*p* = 0.001). The effect size is small (*η*^2^ = 0.01). The percentages of fathers allocated into Clusters 1 to 4 were 12.0%, 34.9%, 22.0%, and 31.1%, respectively (chi-squared = 49.3, *p* < 0.001). The child’s gender did not differ significantly between the four clusters (Table 2). 

The four clusters did not differ in the parental gender ratio except for the parents of the second trimester foetus group (Appendix A). Here, the parental gender ratio was significantly different (*χ*^2^[*df*] = 79.82 [3], *p* < 0.001). Fathers outnumbered mothers in Cluster 2 (Bonding Disorder), whereas mothers outnumbered fathers in Cluster 4 (Positive Bonding). Also, there were statistical differences in all the SPCE mean *Ө* s when compared between the parental gender × child gender (*χ*^2^[*df*] = 296.64 [15], *p* < 0.001) (Appendix A).

*Discriminant function analysis*: The three LDs were calculated to discriminate between the four clusters (Lack of Bonding Emotions, Bonding Disorder, Ambivalent Bonding, and Positive Bonding). The coefficients of linear discriminants are shown in Table 3.

The observed variables in cases can help predict the probability of the cases belonging to certain cluster by the scores of LD1, LD2, and LD3, and the number of cases belonging to the predicted clusters based on the LDA and the true (observed) cluster was counted. For the train data, the sensitivities were 0.841 to 0.996, the specificities were 0.953 to 0.998, the PPVs were 0.900 to 9.984, and the NPVs were 0.947 to 0.997 (Table 4). For the test data, the sensitivities were 0.836 to 0.994, the specificities were 0.951 to 0.998, the PPVs were 0.901 to 0.981, and the NPVs were 0.959 to 0.990. 

## 4. Discussion

In this study, 4600 mothers and fathers were classified into four clusters. These were named Lack of Bonding Emotions, Bonding Disorder, Ambivalent Bonding, and Positive Bonding. The construct validity was shown through associations with the parental bonding emotion typology for the concepts of basic emotions and self-conscious emotions. The Lack of Bonding Emotions Cluster scored low across all the SPCE domains. Parents who belong to this cluster showed little emotion of any kind. They feel almost nothing towards their child. The parents in the Bonding Disorder Cluster showed strong anger, fear, sadness, disgust, shame, and guilt but little positive emotions towards their child. They may emotionally reject or be hostile towards their child. The parents in the Ambivalent Bonding Cluster were characterised by fairly high negative *as well as* positive emotions towards their child. They are likely to shift between the two conflicting emotions: they love, and at the same time, hate their child. The greatest hate springs from the greatest love. The parents in the Positive Bonding Cluster were scored high in positive emotions and low in negative emotions. 

Many previous studies regarding mother-to-infant bonding are described in the context of symptomatology, such as bonding disorders or impaired bonding [41,42,43]. A lack of affection, anger, and rejection in the Japanese version of the Mother-to-Infant Bonding Scale [44,45] and anger and restrictedness, a lack of affection, rejection, and fear in the Postnatal Bonding Questionnaire [46,47] are reported as latent constructs (i.e., syndromes) in factor analyses. The prevalence of a bonding disorder detected by the Japanese version of the Mother-to-Infant Bonding Scale was approximately 12% at one month after childbirth in Japan [48]. In our study, only one out of three parents scored high in positive emotions and low in negative emotions (Positive Bonding). This indicated that the majority of parents had difficult emotions towards their child. 

It was unexpected and surprising that only one third of the parent participants were categorised as *normal*. Does it mean that the remaining parents are unsound and need professional intervention? The term normal has multiple meanings [49]. *Statistical norm* suggests parents with high scores for negative emotion items or low scores for positive emotion items are a majority and therefore *normal*. On the other hand, *value norm* suggests parents with low scores for negative emotion items and high scores for positive emotion items are a minority but healthy (thus, *normal*). In contrast, our results gave us the insight that a majority of parents have difficulty in maintaining a stable affectionate tie with their child. They may struggle in everyday parenting. We do not think of a pathological category of *mental illness* among them (from theoretical consideration as well as lack of taxon for such a condition) but recognise that such parents are targets for professional assessments and supportive (therapeutic) interventions. The four clusters were discriminated by the three LDs that were cross validated using the train and test data sets. The prediction of membership by LDs was fairly accurate. A prediction of the typology may be useful for recognising a target for professional assessment and supportive (therapeutic) intervention in clinical situations.

The four clusters did not differ in terms of the parental age and child gender. On the other hand, different distributions were shown for the parental gender among the four clusters. Different distributions between parental genders were shown in the segments of the second trimester (see Appendix A). Also, distributions for gender combinations of parent–child showed differences among the four clusters (see Appendix A). Within this segment, more fathers were allocated to the Bonding Disorder Cluster than mothers. As compared to a father, a mother might feel stronger emotions about becoming a mother with foetal movements. Rubin [50] described that the tasks of mothering for a woman begin during pregnancy. One of the tasks during pregnancy is building a bond with her child. On the contrary, a father might not feel emotional about becoming a father. A father might feel deprived of a significant other (i.e., a lover) by his own child (foetus). A father’s psychological distress is associated with high levels of using immature ego defences, poorer quality of their current intimate relationship, and poorer social support [51]. Those psychological issues on fathers could lead them to feel negative bonding emotions. Fathers, then, might attain parental roles more slowly than mothers. Hence, we need to promote and induce fathers to attain the parental role as well as mothers. 

Antenatal clinic services are needed to promote parental bonding, and clinicians in obstetric facilities should provide parent-friendly services. A psychological pregnancy review may be effective. Pregnant women who are unhappy about their pregnancy are at risk of psychological adjustment and need specific perinatal mental health care [52]. For both women and men, pregnancy may have a psychological impact on their life, and early intervention may be effective at the time they learn of the pregnancy. Scans (antenatal foetal imaging) are helpful to create a social identity for the unborn baby [53]. For example, medical ultrasound images including three-dimensional (3D) and four-dimensional (4D) ultrasounds could be useful tools to develop parent-to-child emotions for many parents during the antenatal period. Populational approaches are needed to elevate parental bonding conditions during the perinatal period. The father-inclusive perinatal education programmes available to fathers are still few, and the programme effects on father–infant interactions were inconclusive [54]. Effective father-inclusive perinatal parent education programmes are expected to be developed. These interventions may be useful to prevent a child’s maladaptive developmental issues. As mentioned earlier, a higher quality of parental bonding is likely to lead to higher quality child’s attachment, lower colic, easier temperament, and positive mood in the infant.

Generally, the older children become, the more independent from parents they are, and parents grow distant from them. Therefore, emotional conflicts between a parent and a child often occur. Kitamura et al. [44] showed parental bonding difficulties, particularly anger and rejection, were associated with the older age of a child in the observation research targeting parents who have 0- to 10-year-old children. Parents are more likely to experience bonding difficulties as their children get older. In this line, our findings were consistent with those of Kitamura et al. There was a tendency for parents with a school-aged child to be more likely to be classified in the Bonding Disorder Cluster in our findings (see Appendix A). Further research, such as a trajectory focusing on changes in parent-to-child emotions within a person, is expected.

From our findings in this research, four clusters emerged and showed that only one-third of the parent participants were classified as normal. This finding might be brought up by the re-conceptualisation of bonding with the parent-to-child emotions. The targets of many of the past scales to measure ‘parental bonding’ vary considerably, covering not only emotion but also parental identity, uniqueness as a parent, and parental behaviours (including abusive ones). We believe that these concepts should be differentially operationalised with specific measurements [55,56]. Parent-to-child emotions are a useful concept to capture the characteristics of parental bonding.

Our study has several limitations. First, the classification results of this study were limited to a parent to *eldest* child. Different clusters with different characteristics from parents-to-subsequent-child emotions may emerge. We need further investigations of parent-to-child emotions for different populations. Second, we were not yet able to identify matters affecting parent-to-child emotions. Parental norms and gender roles that are relevant to individual value norms may be candidates for those matters [57,58,59]. Third, although we treated SPCE domains as continuous values (dimensional phenomena) measuring each domain of parent-to-child emotions, we did not identify pathological taxa or groups. Because of the cross-sectional nature, this study was unable to identify the trajectory of parental emotions towards a child from pregnancy to childhood [60].

Despite these drawbacks, the typology of the SPCE helps us understand individual differences in terms of parental emotional bonding in clinical situations and research settings.

## 5. Conclusions

We discussed what kinds exists in terms of parent-to-child emotions, what is bonding, and what is normal bonding in this research. Our study findings were that only one out of three parents scored high for positive emotions and low for negative emotions. Antenatal clinic services are needed to promote parental bonding, and clinicians in obstetric facilities should provide parent-friendly services. The typology of the SPCE is helpful for understanding individual differences in terms of parental emotional bonding.

## Figures and Tables

**Figure 1 healthcare-12-00881-f001:**
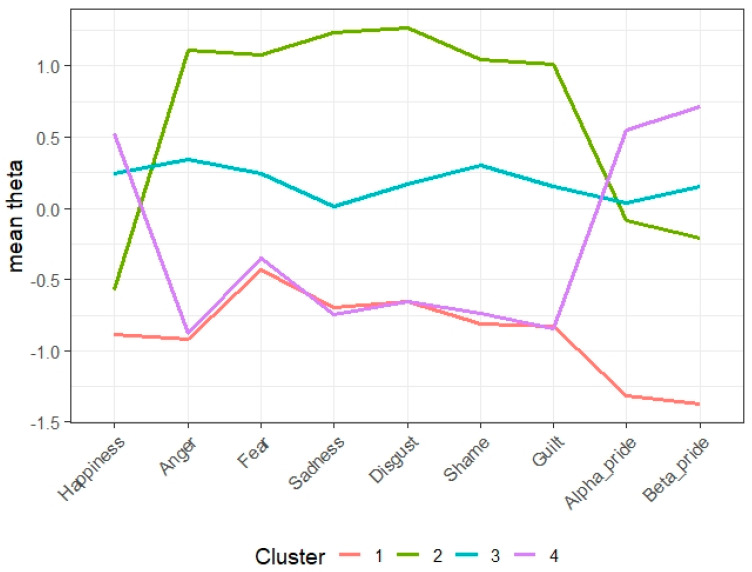
Graph of mean *Ө* s for each SPCE domain by cluster. A solid line coloured pink is Cluster 1 (reduced bonding emotion), a solid line coloured green is Cluster 2 (negative bonding emotion), a solid line coloured blue is Cluster 3 (neutral bonding emotion), and a solid line coloured purple is Cluster 4 (positive bonding emotion).

**Table 1 healthcare-12-00881-t001:** Silhouette analysis of partitioning around medoids (PAM).

	Cluster Sizes and Average Silhouette Widths										
	Cluster 1		Cluster 2		Cluster 3		Cluster 4		Cluster 5		
*k*=	*n*	Silhouette Widths	*n*	Silhouette Widths	*n*	Silhouette Widths	*n*	Silhouette Widths	*n*	Silhouette Widths	Mean Silhouette Widths
1	4600	—	—	—	—	—	—	—	—	—	—
2	2292	0.395	2308	0.383	—	—	—	—	—	—	0.38
3	1291	0.100	1471	0.362	1838	0.276	—	—	—	—	0.25
4	509	0.394	1471	0.351	1211	0.079	1409	0.359	—	—	0.28
5	330	0.480	1471	0.337	1126	0.078	693	0.290	980	0.174	0.24

**Table 2 healthcare-12-00881-t002:** Mean *Ө* s of each SPCE domain by each cluster and the construct validity.

	Cluster 1: Lack of Bonding Emotions	Cluster 2: Bonding Disorder	Cluster 3: Ambivalent Bonding	Cluster 4: Positive Bonding	One Way ANOVA		Tukey Post Hoc Comparison
	*n* = 509 (11.1%)	*n* = 1471 (32.0%)	*n* = 1211 (26.3%)	*n* = 1409 (30.6%)			
	Mean (SD)	Mean (SD)	Mean (SD)	Mean (SD)	*F* (3, 4596)	*η* ^2^	
	Basic emotions						
Happiness	−0.89 (0.94)	−0.58 (0.70)	0.24 (0.69)	0.53 (0.57)	924.62 ***	0.38	1 < 2 < 3 < 4
Anger	−0.92 (0.33)	1.11 (0.64)	0.35 (0.80)	−0.87 (0.38)	3233.88 ***	0.68	1, 4 < 3 < 2
Fear	−0.43 (0.49)	1.08 (0.51)	0.25 (0.69)	−0.35 (0.49)	1887.49 ***	0.54	1, 4 < 3 < 2
Sadness	−0.69 (0.34)	1.23 (0.53)	0.01 (0.51)	−0.75 (0.23)	5729.21 ***	0.79	4, 1 < 3 < 2
Disgust	−0.65 (0.40)	1.27 (0.56)	0.17 (0.53)	−0.66 (0.32)	4614.07 ***	0.75	4, 1 < 3 < 2

Shame	−0.81 (0.35)	1.05 (0.52)	0.30 (0.49)	−0.74 (0.36)	4558.60 ***	0.75	1, 4 < 3 < 2
Guilt	−0.83 (0.35)	1.01 (0.50)	0.15 (0.59)	−0.85 (0.29)	4778.55 ***	0.76	4, 1 < 3 < 2
Alpha pride	−1.32 (0.53)	−0.09 (0.66)	0.04 (0.75)	0.55 (0.80)	856.45 ***	0.36	1 < 2 < 3 < 4
Beta pride	−1.37 (0.65)	−0.21 (0.62)	0.15 (0.71)	0.71 (0.73)	1260.95 ***	0.45	1 < 2 < 3 < 4
Parents’ age	35.5 (6.73)	36.9 (7.24)	35.9 (6.51)	35.5 (6.75)	12.23 **	0.01	1, 4 < 3, 2

	*n* (%)	*n* (%)	*n* (%)	*n* (%)	Total *n* (%)		*χ*^2^ (*df*)
Parental gender							49.3 (3) ***
Father	279 (12.0)	815 (34.9)	515 (22.0)	727 (31.1)	2336 (100.0)		
Mother	230 (10.2)	656 (29.0)	696 (30.7)	682 (30.1)	2264 (100.0)		
Childs’ gender							20.7 (6) **
Boy	230 (10.2)	740 (32.8)	610 (27.0)	678 (30.0)	2258 (100.0)		
Girl	242 (11.6)	632 (30.3)	558 (26.7)	654 (31.4)	2086 (100.0)		
Unknown	37 (14.4)	99 (38.7)	43 (16.8)	77 (30.1)	256 (100.0)		

Note:**, *p* < 0.01; ***, *p* < 0.001

**Table 3 healthcare-12-00881-t003:** Linear discriminant analysis (LDA) (train data: *n* = 2300).

	LD1	LD2	LD3
Coefficients of linear discriminants			
Happiness	−0.464	0.380	−0.459
Anger	0.595	−0.074	−0.843
Fear	0.454	−0.118	−0.431
Sadness	0.723	−0.005	1.558
Disgust	0.362	0.137	0.712
Shame	0.286	0.508	−0.764
Guilt	0.740	−0.010	−0.480
Alpha pride	0.009	0.603	0.014
Beta pride	0.157	0.750	0.539

**Table 4 healthcare-12-00881-t004:** The number of cases belonging to predicted clusters based on the LDA and true (observed) clusters, sensitivity, specificity, positive predictive value (PPV), and negative predictive value (NPV).

*Train data (n = 2300)*					
		True (observed) cluster			
		1:	2:	3:	4:
Predicted cluster	1:	240	0	3	1
	2:	0	744	31	0
	3:	10	3	501	4
	4:	15	0	61	687
Sensitivity		0.906	0.996	0.841	0.993
Specificity		0.998	0.980	0.990	0.953
PPV		0.984	0.960	0.967	0.900
NPV		0.988	0.998	0.947	0.997
*Test data (n = 2300)*					
		True (observed) cluster			
		1:	2:	3:	4:
Predicted cluster	1:	203	1	5	35
	2:	0	717	7	0
	3:	9	328	531	47
	4:	1	0	4	712
Sensitivity		0.836	0.987	0.884	0.994
Specificity		0.998	0.986	0.988	0.951
PPV		0.981	0.970	0.965	0.901
NPV		0.981	0.990	0.959	0.997

Note: Train data (*n* = 2300), the data for carrying out LDA; test data (*n* = 2300), the data for validating LDA.

## Data Availability

The data set analysed and used in this study is available upon reasonable request to the first author.

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
