# Peer review of "Typology of Parent-to-Child Emotions: A Study of Japanese Parents of a Foetus up to a 12-Year-Old Child"

_healthcare, 2024, doi:10.3390/healthcare12090881_

Round 1
Reviewer 1 Report
Comments and Suggestions for Authors
Thank you for your contribution to the literature in the area of parent/child emotions. This manuscript provides data regarding types of parental (both maternal and paternal) emotions toward their eldest child (ages ranging from prenatal to 12 years). Four types of parent-to-child emotions were identified in this cluster analysis.
The methods and results are described clearly. The discussion is thoughtful, including implications, limitations, and acknowledgements about the study examining only parents' emotions toward their eldest child, and a discussion about the term 'normal.'
The introduction is thorough in describing various emotions and how they arise in parenthood. The Scale for Parent-to-Child Emotions is also explained. The only recommendation at this time is to include something about attachment in the introduction. The term 'bonding' is often associated with attachment, and there is a wealth of literature regarding attachment that can/should be incorporated (even if briefly) into this introduction.
Author Response
Reply to reviewers’ comments:
We thank all the reviewers for the helpful comments for correction. We have taken the comments on board to improve and clarify the manuscript. We very much hope the revised manuscript is accepted for publication in Healthcare.
Reviewer: 1
Comments and Suggestions for Authors
Thank you for your contribution to the literature in the area of parent/child emotions. This manuscript provides data regarding types of parental (both maternal and paternal) emotions toward their eldest child (ages ranging from prenatal to 12 years). Four types of parent-to-child emotions were identified in this cluster analysis.
The methods and results are described clearly. The discussion is thoughtful, including implications, limitations, and acknowledgements about the study examining only parents' emotions toward their eldest child, and a discussion about the term 'normal.'
The introduction is thorough in describing various emotions and how they arise in parenthood. The Scale for Parent-to-Child Emotions is also explained. The only recommendation at this time is to include something about attachment in the introduction. The term 'bonding' is often associated with attachment, and there is a wealth of literature regarding attachment that can/should be incorporated (even if briefly) into this introduction.
We inserted sentences into introduction section and revised as follows:
Although the terms ‘attachment’ and ‘bonding’ are sometimes used interchangeably, they should be distinguished. The term ‘bonding’ has the direction of a parent to a child/foetus. On the other hand, the term ‘attachment’ has the direction of a child/foetus to a parent. Walsh [7] pointed out that the terminology of attachment is not accurate in describing how parents feel. Furthermore, Kinsey and Hupcey [8] proposed that the affective state of the mother; maternal feelings and emotions towards the infant are the primary indicator of maternal infant bonding. Kinsey and Hupcey [6] defined maternal-infant bonding as feelings and emotions towards infants. Capturing parent-to-child emotions by basic and self-conscious emotions should be the basis of the concept of bonding [6].
(Introduction, Line 56-65)

Reviewer 2 Report
Comments and Suggestions for Authors
Dear authors,
The manuscript addresses a relevant issue and provides sufficient background and relevant references. The conclusions are well supported by the data. However, the text presented in lines 123 to 127 could be better integrated with the previous text.
Overall, the manuscript publication is recommended.
Author Response
Reply to reviewers’ comments:
We thank all the reviewers for the helpful comments for correction. We have taken the comments on board to improve and clarify the manuscript. We very much hope the revised manuscript is accepted for publication in Healthcare.
Reviewer: 2
Comments and Suggestions for Authors
Dear authors
The manuscript addresses a relevant issue and provides sufficient background and relevant references. The conclusions are well supported by the data. However, the text presented in lines 123 to 127 could be better integrated with the previous text.
(The reviewer suggested that the lines 123 to 127 [shown in the text with blue highlight] should be integrated with the previous text. This seems curious to us. We may not fully understand what he/she meant to say.) Following the reviewer’s comment, we inserted the sentence as follows:
The SPCE may facilitate several clinical or research questions regarding parental emotional bonding. The SPCE is likely to make individual differences in parental emotional bonding more apparent.
What remained to be answered was how many types (groups) parents can be categorised into in terms of parental emotional bonding states: what kinds of characteristics identify parents of these groups of parental emotional bonding. The latter includes: Do fathers and mothers differ in such categories? Can we estimate groups from the SPCE scores?
Overall, the manuscript publication is recommended.

Reviewer 3 Report
Comments and Suggestions for Authors
Thankyou for such a careful and detailed research paper. I can see that you have used bonding extensively in the paper, and obviously the paper is about bonding and emotions. However, I would like to have seen a discussion about the important differences between attachment and bonding and the relative significance of to the field of study in which your paper sits. I would also like to have seen some tentative implications for further work in relation to emotion ratings and quantitative analysis at parent bonding, and child attachment.
Author Response
Reply to reviewers’ comments:
We thank all the reviewers for the helpful comments for correction. We have taken the comments on board to improve and clarify the manuscript. We very much hope the revised manuscript is accepted for publication in Healthcare.
Reviewer: 3
Comments and Suggestions for Authors
Thankyou for such a careful and detailed research paper. I can see that you have used bonding extensively in the paper, and obviously the paper is about bonding and emotions. However, I would like to have seen a discussion about the important differences between attachment and bonding and the relative significance of to the field of study in which your paper sits. I would also like to have seen some tentative implications for further work in relation to emotion ratings and quantitative analysis at parent bonding, and child attachment.
As we responded above (to reviewer 1), we inserted sentences into introduction section (Line 56-65). Also, we discussed as follows:
Our findings in this research, four clusters emerged and showed that only one-third of the participant parents were classified as normal. This finding might be brought up by the re-conceptualisation of bonding with the parent-to-child emotions. The targets of many of the past scales to measure ‘parental bonding’ vary considerably covering not only emotion but also parental identity, uniqueness as a parent, and parental behavious (including abusive ones). We believe that these concepts should be differentially operationalised with specific measurements [56-57]. Parent-to-child emotions are a useful concept to capture characteristics of parental bonding.
(Discussion, Line 380-387)

Reviewer 4 Report
Comments and Suggestions for Authors
Dear authors,
Congratulations for the presented work. It is very interesting, and the type of analysis developed can have impact in interventions and research regarding parent-to-child emotions.
Just some comments to improve your work:
Lines 103, 113, 190, 327 – check references.
It would be interesting to identify other instruments, in addition to the SPCE scale, available for this type of assessment and the advantages of the scale used.
The justification of the method and analyzes could be inserted in chapter 2.
Although it is clear that the description of the sample has already been carried out in other work, it would be interesting to include a brief description in this study.
In the analysis procedures, when ANOVAS and Chi-Squares are indicated, it would be relevant to identify the significance level used (e.g., 5%), as well as calculate the effect size.
In the results, the cutoff criteria of the 5th cluster is not clear.
Line 254 - Is the result not significant?
Discussion could be enrich regarding the clusters' features and other studies, was well as indications toward future interventions.
Author Response
Reply to reviewers’ comments:
We thank all the reviewers for the helpful comments for correction. We have taken the comments on board to improve and clarify the manuscript. We very much hope the revised manuscript is accepted for publication in Healthcare.
Reviewer: 4
Comments and Suggestions for Authors
Dear authors,
Congratulations for the presented work. It is very interesting, and the type of analysis developed can have impact in interventions and research regarding parent-to-child emotions.
Just some comments to improve your work:
Lines 103, 113, 190, 327 – check references.
We have corrected it following comments from the reviewers.
It would be interesting to identify other instruments, in addition to the SPCE scale, available for this type of assessment and the advantages of the scale used.
The justification of the method and analyzes could be inserted in chapter 2.
Although it is clear that the description of the sample has already been carried out in other work, it would be interesting to include a brief description in this study.
In the analysis procedures, when ANOVAS and Chi-Squares are indicated, it would be relevant to identify the significance level used (e.g., 5%), as well as calculate the effect size.
Following the suggestion of the reviewer, we have included effect sizes in Table 2. Effect size’s results were also added in the text.
In the results, the cutoff criteria of the 5th cluster is not clear.
We used silhouette withs (silhouette indices) to determine cluster numbers (See, Line 217-218, yellow highlighted). The second-best mean silhouette width was 0.28 (k = 4), which was better than 5-cluster solution. We also considered interpretability to clinical situations (See, Line 244-248, yellow highlighted).
Line 254 - Is the result not significant?
We have carefully reviewed the results of our analysis and have revised them as follows:
Because of multiple comparison, we set significant level at 0.1% (p < .001).
(inserted to 2.3. Statistical Analyses, Line 230)
The parents’ mean age comparisons between the four clusters showed that cluster 2 was significantly higher than clusters 1 and 4 (p<.001), however, not significant in comparison with cluster 3 (p = .001). The effect size is small (η2 = .01).
(Results, Line 266-268)
Discussion could be enrich regarding the clusters' features and other studies, was well as indications toward future interventions.
As we responded above (to the reviewer 1 and 2), we inserted sentences into introduction section (Line 55-64). Also, we discussed as follows,
Our findings in this research, four clusters emerged and showed that only one-third of the participant parents were classified as normal. This finding might be brought up by the re-conceptualisation of bonding with the parent-to-child emotions. The targets of many of the past scales to measure ‘parental bonding’ vary considerably covering not only emotion but also parental identity, uniqueness as a parent, and parental behavious (including abusive ones). We believe that these concepts should be differentially operationalised with specific measurements [56-57]. Parent-to-child emotions are a useful concept to capture characteristics of parental bonding.
(Discussion, Line 380-387)

Reviewer 5 Report
Comments and Suggestions for Authors
This is an ambitious and very complex research topic. In my opinion, knowing the forms of manifestation and the possibilities of integration of the child in the context of the typology of parent-to-child emotions is something that could contribute a lot to the educational process. Based on these considerations I would offer some suggestions:
- I think it would be necessary to specify more clearly the perspective in which the results of this study will be used. One can speak of a medical perspective if the early framing of emotional relationships between parents and children can be considered as a risk in the later development of psychological or psychiatric disorders. On the other hand, an educational perspective can be significant if monitoring of emotional relationships is used as a form of establishing educational needs.
- From the point of view of content, from my observations, it is to be avoided that results and conclusions are written in the abstract. Here I recommend to focus the abstract on objectives, hypotheses and research methodology.
- I suggest an additional detailing of the measurement scales, even a presentation of the questions used to measure basic emotions and self-conscious emotions.
- The authors stated that it was unable to identify the trajectory of parental emotion towards a child from pregnancy to childhood. On the other hand, the sample was constructed as 20 segments: two parental genders * ten age ranges of children. This context allows two perspectives of analysis that which can be found at the end of the article as supplementary table 1 and 2. My opinion is that the two tables should be included in the paper and interpreted, because they represent two perspectives that should not be ignored:
1. A comparative analysis of the trends in parent-child emotional bonding in terms of the gender of the parents (which determines the role of mother or father).
2. An analysis of parental emotional bonding according to age ranges of children. The structure of parent-child emotional bonding is likely to differ significantly according to the age range of the child.
- I also recommend the introduction of a conclusion chapter.
Author Response
Reviewer: 5
Comments and Suggestions for Authors
This is an ambitious and very complex research topic. In my opinion, knowing the forms of manifestation and the possibilities of integration of the child in the context of the typology of parent-to-child emotions is something that could contribute a lot to the educational process. Based on these considerations I would offer some suggestions:
- I think it would be necessary to specify more clearly the perspective in which the results of this study will be used. One can speak of a medical perspective if the early framing of emotional relationships between parents and children can be considered as a risk in the later development of psychological or psychiatric disorders. On the other hand, an educational perspective can be significant if monitoring of emotional relationships is used as a form of establishing educational needs.
We inserted sentences in Introduction section and Discussion section as follows:
This is the case for parent-to-child emotions. A parent feels various emotions that motivate adaptive behaviours in the interaction of the parent-child dyad. For example, higher quality of maternal bonding was correlated with higher quality of the child’s attachment, attachment to the parent, parent‐reported lower colic rating, easier temperament, and positive infant mood [2]. The emotions a parent feels towards a child play an important role in the quality of their interaction. Therefore, it is meaningful to clarify what kind of emotions parents feels within interaction between parent and child.
(Introduction, Line, 43-48)
Effective father-inclusive perinatal parent education programs are expected to be developed. Those interventions may be useful to prevent a child’s maladaptive developmental issues. As mentioned earlier, a higher quality of parental bonding is likely to lead to higher quality child’s attachment, lower colic, easier temperament, and positive mood in the infant.
(Discussion, Line, 366-369)
- From the point of view of content, from my observations, it is to be avoided that results and conclusions are written in the abstract. Here I recommend to focus the abstract on objectives, hypotheses and research methodology.
We are sorry about that but Journal requires a structured abstract in which results and conclusions are mandatory. We would like to leave this issue up to the Editor.
- I suggest an additional detailing of the measurement scales, even a presentation of the questions used to measure basic emotions and self-conscious emotions.
In order to rigorously protect our copy right, we should like to avoid the disclosure of the item-by-item. We are afraid that description such as ‘Prohibit the conversion or use without authorization’ are not complete to protect from illegitimate use or copying.
- The authors stated that it was unable to identify the trajectory of parental emotion towards a child from pregnancy to childhood. On the other hand, the sample was constructed as 20 segments: two parental genders * ten age ranges of children. This context allows two perspectives of analysis that which can be found at the end of the article as supplementary table 1 and 2. My opinion is that the two tables should be included in the paper and interpreted, because they represent two perspectives that should not be ignored:
- A comparative analysis of the trends in parent-child emotional bonding in terms of the gender of the parents (which determines the role of mother or father).
- An analysis of parental emotional bonding according to age ranges of children. The structure of parent-child emotional bonding is likely to differ significantly according to the age range of the child.
As in the reviewer's comments, Supplementary Tables include important perspectives of parental emotions towards a child from pregnancy to childhood, and these are also our interests. However, it would devote a lot of space to include in the main stream of this paper. So, we would like to keep these tables as supplementary. In addition, a series of comparative analysis of the trends in parent-child emotional bonding in terms of the gender of the parents and age range of children can be interpreted that there were NOT significant differences. Because chi-squared tests provide only information about different distributions of all groups, it is not known which groups have significant differences in the observed values. Although post hoc test is recommended to find the source of overall significance, huge number of times of tests in combinations of groups. We expect to perform in the future research. Discussions about differences in the parent’s gender and combinations of parent-child have already been stated (shown with yellow highlighted), we also inserted following sentences:
Generaly, the older children become, the more independent on parents they are, and parents get away from them. Therefore, emotional conflicts between a parent and a child often occur. Kitamura et al. [55] showed parental bonding difficulties, particularly Anger and Rejection, were associated with the older age of the child in the observation research targeting parents who have 0 to 10 old children. Parents are more likely to experience bonding difficulties as their children gets older. In this line, our findings were consistent with those of Kitamura et al. There was a tendency for parents with a school-age child to be more likely to be classified in the Bonding disorder cluster in our findings (See, Supplementary Table 1). Further research such as a trajectory focusing on changes in parent-to-child emotions within a person, is expected.
(Discussion Line 370-379)
- I also recommend the introduction of a conclusion chapter.
We inserted Conclusions chapter (Line 400-).
